# Terpinen-4-ol, the Main Bioactive Component of Tea Tree Oil, as an Innovative Antimicrobial Agent against *Legionella pneumophila*

**DOI:** 10.3390/pathogens11060682

**Published:** 2022-06-14

**Authors:** Francesca Mondello, Stefano Fontana, Maria Scaturro, Antonietta Girolamo, Marisa Colone, Annarita Stringaro, Maura Di Vito, Maria Luisa Ricci

**Affiliations:** 1Department of Infectious Diseases, Istituto Superiore di Sanità, Viale Regina Elena, 299, 00161 Rome, Italy; stefano.fontana@iss.it (S.F.); maria.scaturro@iss.it (M.S.); antonietta.girolamo@iss.it (A.G.); 2Società Italiana per la Ricerca sugli Oli Essenziali—SIROE (Italian Society for Research on Essential Oils), Viale Regina Elena, 299, 00161 Rome, Italy; annarita.stringaro@iss.it; 3ESCMID Study Group for *Legionella* Infections (ESGLI), 4001 Basel, Switzerland; 4National Center for Drug Research and Evaluation, Istituto Superiore di Sanità, Viale Regina Elena, 299, 00161 Rome, Italy; marisa.colone@iss.it; 5Dipartimento di Scienze Biotecnologiche di Base, Cliniche Intensivologiche e Perioperatorie, Università Cattolica del Sacro Cuore, Largo A. Gemelli 8, 00168 Rome, Italy

**Keywords:** essential oil, *Legionella pneumophila*, tea tree oil, terpinen-4-ol, vapors, antibacterial activity, broth micro-dilution method, micro-atmosphere diffusion method, time killing, scanning electron microscopy (SEM)

## Abstract

*Legionella pneumophila* (*Lp*), responsible for a severe pneumonia called Legionnaires’ disease, represents an important health burden in Europe. Prevention and control of *Lp* contamination in warm water systems is still a great challenge often due to the failure in disinfection procedures. The aim of this study was to evaluate the in vitro activity of Terpinen-4-ol (T-4-ol) as potential agent for *Lp* control, in comparison with the essential oil of *Melaleuca alternifolia* (tea tree) (TTO. Minimum Inhibitory Concentration (MIC) and Minimum Bactericidal Concentration (MBC) of T-4-ol were determined by broth micro-dilution and a micro-atmosphere diffusion method to investigate the anti-*Lp* effects of T-4-ol and TTO vapors. Scanning Electron Microscopy (SEM) was adopted to highlight the morphological changes and *Lp* damage following T-4-ol and TTO treatments. The greatest antimicrobial activity against *Lp* was shown by T-4-ol with a MIC range of 0.06–0.125% *v*/*v* and MBC range of 0.25–0.5% *v*/*v*. The TTO and T-4-ol MIC and MBC decreased with increasing temperature (36 °C to 45 ± 1 °C), and temperature also significantly influenced the efficacy of TTO and T-4-ol vapors. The time-killing assay showed an exponential trend of T-4-ol bactericidal activity at 0.5% *v*/*v* against *Lp*. SEM observations revealed a concentration- and temperature- dependent effect of T-4-ol and TTO on cell surface morphology with alterations. These findings suggest that T-4-ol is active against *Lp* and further studies may address the potential effectiveness of T-4-ol for control of water systems.

## 1. Introduction

*Legionella*, an opportunistic Gram-negative bacterium, ubiquitously found in freshwater environments and artificial water systems, represents a serious public health problem, mainly for elderly and immunocompromised people. The genus *Legionella* currently consists of 63 species, of which *Legionella pneumophila* (*Lp*) is responsible for most human infections [1]. *Lp* can be distinguished into 15 serogroups, of which *Lp* serogroup 1 is the most virulent and associated with over 80% of Legionnaires’ disease (LD) cases in Europe [1]. The infection occurs through inhalation or aspiration of *Lp*-contaminated aerosols and usually gives rise to two clinical syndromes: a life-threatening pneumonia called LD or Pontiac fever, a self-limited flu-like illness [2,3].

Since 2008, when LD was first reported to The European Surveillance System (Tessy), coordinated by the European Centre for Disease Prevention and Control (ECDC), an increasing trend (with the exception and unexplained decrease in 2011) has been observed, reaching a peak of 11,405 LD cases in 2018 (23% more cases than in 2017). This number remained unchanged in 2019 but decreased in 2020 following the Covid-19 pandemic [4,5]. The same trend has been observed in Italy, as incidence rates per million of population increased from 16.1 in 2011 to 52.9 in 2019, with a 35% notable decrease in 2020 [6,7,8]. Community and travel-associated outbreaks of LD have been mostly linked with cooling towers, building water-distribution systems, spas, fountains, and public baths. Water-distribution systems and aerosol-producing medical devices are main sources of health care-associated legionellosis [9]. Person-to-person transmission of LD is exceptional [10], so prevention control measures have focused on eliminating the pathogen from water systems. To date, the long-term control of *Legionella* contamination of water distribution and air conditioning systems still represents a great challenge, to both health and the structural complexity of the water systems [1]. *Lp* proliferate inside free-living protozoa [11] and colonize multispecies biofilms [12,13] that allow persistence in water systems even when high temperature, biocide treatment, scale, and corrosion are control measures. Therefore, the development of alternative or integrative methods that can counteract and prevent the colonization of *Legionella* in water distribution systems and minimize exposure to potentially hazardous chemicals is considered a high priority. Of all the pathogens present in water, *Legionella* is the one that causes the greatest health burden in the EU [14]. Recently, antimicrobial agents extracted from a variety of plants have been evaluated in particular essential oils (EOs) [15,16,17,18].

The biological activities of EOs are considered to have preventive and therapeutic benefits [19,20,21]. The antimicrobial activity of EOs and their components has been extensively investigated, both in aqueous and in vapor phases [22,23,24], against bacteria, fungi, viruses, parasites, and insects, resulting in the formulation of patents [25] and applications in the pharmaceutical technology [26]. In 2009, we pioneered in vitro bactericidal efficacy of the essential oil of *Melaleuca alternifolia* (tea tree) (TTO) against *Lp* strains by improving the susceptibility method with vapor control of TTO [27].

To our knowledge, studies on the in vitro activity of Terpinen-4-ol (T-4-ol), the main bioactive component of TTO, against *Lp*, are lacking. The aim of this study was to evaluate the in vitro activity of T-4-ol, both in the aqueous and the in vapor phase, against *Lp* sg1 and sg6 human and environmental isolates, proven to be among the most widespread in human infection in Italy and Europe [1,28].

This is the first report that highlights the in vitro susceptibility of *Lp* to T-4-ol in comparison to the anti-*Lp* activity of TTO, with the aim of developing new disinfectants for contamination control of *Legionella* in water distribution systems.

## 2. Results

### 2.1. Comparison of Anti-Legionella Activities of Terpinen-4-ol and Tea Tree Oil Analyzed by Broth Micro-Dilution Method

We compared the activities of T-4-ol and TTO against all the different strains of *Lp* sg1 and sg6 using the broth micro-dilution method with sterilized Transparent Microplate Sealer (TMS) at 36 ± 1 °C. This method was used to exclude vapor diffusion spreading over the wells, responsible for inter-experimental variation of Minimum Inhibitory Concentration (MIC) and Minimum Bactericidal Concentration (MBC), as described previously [27]. *Lp* was sensitive to T-4-ol, with MIC ranging from 0.06 to 0.125% *v*/*v*, and MBC at 0.5% *v*/*v*, while TTO MIC and MBC ranging from 0.125 to 0.5% *v*/*v*, and 0.5–1.0% *v*/*v* respectively (Table 1).

The MIC and MBC average values of T-4-ol and TTO against the two *Lp* serogroups were compared (Table 2). The uniformity of MIC values at 0.25% *v*/*v* for TTO and at 0.125% *v*/*v* for T-4-ol are noticeable, with rare exceptions, while the MBC values at 0.5% *v*/*v* was the same for both TTO and for the main component T-4-ol in both tables (Table 1 and Table 2). No difference either between the two serogroups or among the reference strains environmental and clinical isolates was seen, and for this reason the strains were grouped according to their serogroup in Table 2.

Figure 1 shows the comparison of T-4-ol and TTO inhibitory activities at temperatures of 36 °C, 40 °C, and 45 ± 1 °C against six strains of *Lp* sg1 and sg6 by broth micro-dilution method. MIC values of T-4-ol were always lower than the corresponding MICs obtained by testing the TTO. Furthermore, MICs of T-4-ol and TTO decreased with increasing temperature. This modulation was more evident for TTO, where the temperature significantly improved the MIC values (*p* < 0.0002). Under the same conditions, the MIC values obtained when testing T-4-ol at 36 °C and 40 ± 1 °C were always statistically more significant than those obtained with TTO (*p*_T-4-ol_ < 0.0001 and *p*_TTO_ < 0.0002, respectively).

The comparison of TTO and T-4-ol bactericidal activities (MBC) at temperatures of 36 °C, 40 °C and 45 ± 1 °C against six strains of *Lp* serogroups 1 and 6 with broth micro-dilution method was showed in Figure 2.

The temperature-dependent trend described for the MIC variation was also observed for MBCs. The MBC values with T-4-ol were lower than the correspondent values obtained with TTO. Furthermore, for both the natural products tested, the MBCs at 45 °C showed statistically significant differences to both 36 ± 1 °C (*p* < 0.002) and 40 °C (*p*_T-4-ol_ < 0.002 and *p*_TTO_ < 0.03, respectively).

### 2.2. Comparison of Anti-Legionella Activities of Terpinen-4-ol and Tea Tree Oil Analyzed by the Micro-Atmosphere Diffusion Method

The anti-*Lp* activities of T-4-ol and TTO were also tested with a micro-atmosphere diffusion method, which allowed the contact of only T-4-ol or TTO vapors with bacterial cells at different temperatures. The same six selected *Lp* strains, examined by the broth micro-dilution method, were tested at 36 °C and 45 ± 1 °C to evaluate the antimicrobial effects of T-4-ol and TTO vapors. With this method, the vapor originating from T-4-ol caused total growth inhibition, as measured after 2 days of culture, with total killing after 7 days of culture. The results of the anti-*Lp* activities of the T-4-ol and TTO vapors are shown in Table 3 and Figure 3.

The comparison of the anti-*Lp* activities of T-4-ol and TTO vapors at temperatures of 36 °C and 45 ± 1 °C for the selected strains of *Lp* sg1 (one reference strain ATCC 33152, one clinical strain and one environmental strain) and *Lp* sg6 (one reference strain ATCC 33215, one clinical strain and one environmental strain) after 7 days of incubation are shown in Table 3. The diameter of the inhibition of the tested *Lp* strains was variable and was sometimes equal to the internal diameter of the Petri dish (90 mm). The *Lp* sg1 ATCC 33152 strain did not grow at 45 °C therefore it was not considered for experiments at that temperature [29]. The T-4-ol/TTO ratio was always >1, which mean that the average of the inhibition halos obtained with T-4-ol was always greater than that obtained with TTO at 36 ± 1 °C. The T-4-ol/TTO ratio was instead ≥1 at 45 ± 1 °C.

### 2.3. Anti-Legionella pneumophila Activity of Terpinen-4-ol Analyzed by Time-Killing Test

In the time-killing experiments, the in vitro survival of *Legionella* as a function of T-4-ol concentration and contact times was investigated (Figure 3). We highlighted the exponential trend of T-4-ol bactericidal activity against *Lp*. The vitality of *Lp* decreased over time in the presence of a concentration of 0.5% *v*/*v* of T-4-ol (corresponding to the MBC value). After 5 min, the mortality percentage was 54.3% ± 3.2%, while after 10 min, the mortality was 100%. The graph shows a polynomial decrease of *Legionella* vitality as a function of time (R = 0.921). The t-test indicated that the activity of T-4-ol at a concentration of 0.5% *v*/*v* was statistically significant (*p* = 0.0136).

### 2.4. Morphological Effects of Terpinen-4-ol and Tea Tree Oil on Legionella pneumophila Analyzed by Scanning Electron Microscopy (SEM) at Different Temperatures

SEM analyses performed on untreated bacteria (controls) at 30 °C, 40 °C, and 45 °C revealed the typical rod shape and that temperature had a no effect on *Lp* morphology. Most cells were 2–10 µm long and the *Lp* surface appeared slightly wrinkled.

Figure 4 shows the effect of treatments (1%, 2.5% *v*/*v* of TTO and 1% *v*/*v* of T-4-ol) at 40 °C. Specifically, in Figure 4B (1% *v*/*v* TTO treatment) the surface morphology appears more wrinkled than control cells. Exposure to T-4-ol (0.42% *v*/*v*) induced numerous small bleb formations on the bacterial cell surface. When the highest (2.5% *v*/*v* TTO and 1% *v*/*v* T-4-ol) concentrations were used, *Lp* modifications were more evident. Figure 4D shows that 2.5% *v*/*v* TTO caused bleb formations and the cell surface was more rugose, with some cells appearing to swell at the center. Openings were present on surface, probably as a result of the inner membrane rupturing. Moreover, numerous blebs were present on the coverslips (Figure 4D). Figure 4E shows that treatment with 1% *v*/*v* T-4-ol induced the swelling and wrinkling of the cell surface. In addition, cells had a spheroplast-like collapsed structure.

Figure 5 shows the effect of the same treatments at 45 °C. In particular, Figure 5A shows bacteria at 45 °C temperature-exposed: cells appear with their typical rod shape without any ultrastructural modification due to the increase in temperature. In Figure 5B (1% *v*/*v* TTO) the cell surface appears more wrinkled with mid-cell swellings visible (arrowheads). Blebs are also present on the bacteria surface and on coverslips (arrowheads). Figure 5C shows multiple damages to the cell that have merged (arrow) with outer cell wall and inner membrane ruptures (arrowheads). TTO at a concentration of 2.5% *v*/*v* induced numerous blebs that are visible on numerous rugose cell surfaces (arrows) (Figure 5D). T-4-ol at a concentration of 1% *v*/*v* produced collapsed cells (Figure 5E). The cell surfaces appeared wrinkled, but their organization was different from the cell surface shown in Figure 5C.

## 3. Discussion

Investigations on the antimicrobial activity of EOs on *Lp* are limited in number and have only circumstantially examined *Lp* while they were in a context of EOs activity against multiple other bacteria [30,31,32,33,34,35,36,37,38].

*Melaleuca alternifolia* (Maiden & Betch) Cheel (tea tree) EO (TTO) has been studied for its antimicrobial activity, including in the vapor phase [39], against a large number of Gram-negative and -positive bacteria and fungi by in vitro, preclinical and clinical studies [22,40,41]. TTO contains more than 100 components, such as monoterpenes, sesquiterpenes, and phenolic compounds. The most abundant compound is T-4-ol, in an amount of at least 30% [40], but is also present in many EOs from other plant species. Various activities of T-4-ol have been described in the literature, encompassing antibacterial, antifungal, antiviral, and insecticidal effects, as well as antitumoral and anti-inflammatory activity [42,43,44,45,46,47]. Furthermore, it was highlighted that T-4-ol is the principal component responsible for the antimicrobial activity of TTO, as both possess similar antimicrobial effects [48,49]. Regarding the activity of T-4-ol against *Lp*, there has only been one study [33]. This study investigated the antibacterial activity of a fraction containing 53% of T-4-ol obtained from EO of *Chamaecyparis obtusa.* The authors showed high bacteriostatic activity against all Gram-positive and -negative tested strains, including a single *Lp* strain. However, the fraction was made up of many monoterpenes and sesquiterpenes that probably contributed to T-4-ol antibacterial activity. Pure synthetic T-4-ol has been tested against a number of bacterial strains but not *Lp.* Therefore, it was not possible to draw conclusions on its sole efficacy against *Legionella* in that study [33].

Our study focused attention on the main component of TTO, T-4-ol, highlighting the in vitro activity against *Lp*, in the aqueous and vapor phase, as we had previously shown for TTO [27]. In addition, the anti-*Lp* activity of T-4-ol was tested in comparison with TTO to verify to what extent the antibacterial phytocomplex activity could be influenced by its main component (42.35% of the mixture), at various temperatures.

Various types of non-standardized methods have already been reported in the literature for an assessment of the antimicrobial activity of EOs and their components, resulting in data variation [50]. Van de Vel and collaborators [50] pointed out that the different parameters used in these techniques (the incubation conditions, the culture media, and the use of emulsifiers/solvents) influenced the Minimum Inhibitory Concentration (MIC), causing this large variance.

EOs are partially hydrophobic substances with volatile components, therefore it is important to standardize a method to evaluate the antimicrobial activity of these oils, particularly considering the vapor emission effects seen in the aqueous phase [51,52]. There are a few publications in the literature [34,39,53,54], but no standardized method for testing EO vapors has been reported [20,55]. A meaningful comparison of the different studies remains, at best, problematic [56,57,58]. The parallel evaluation of the antimicrobial activity of EOs using standardized methods both in the aqueous phase, with evaporation control, and in the vapor phase, should provide more reliable and comparable results. It should be noted that some EOs may have peculiar characteristics in the liquid or vapor phase that could lead to the expression of different biological activities.

All the existing studies on the antimicrobial activity of EOs against *Legionella* have tested natural products with chemical compositions that often did not comply with the regulations or standardized methods. These all affect the anti-*Legionella* activity of the EOs and limit comparisons to determine the potential role of EOs in controlling the growth of the *Legionella*. Furthermore, only one or few *Legionella* strains have previously been tested against EOs [30,31,32,33,34,35].

The only study on the anti-*Legionella* activity of TTO, both in the liquid and in the vapor phases, dates back to our group [27]. To date, the in vitro activity of T-4-ol against quite a number of *Lp* strains, has never been tested, and this is the first report on the in vitro susceptibility of *Lp* to T-4-ol, both in the liquid and in the vapor phase, in comparison with anti-*Lp* activity of TTO.

To this aim, we tested the bacteriostatic and bactericidal activities of T-4-ol and TTO, compliant with international and European regulations, against 21 clinical and environmental *Lp* strains by an established standard broth micro-dilution (BMD), with slight modifications, and a micro-atmosphere diffusion method previously reported [27].

In the BMD method, microplate sealing was considered as previous work highlighted EO evaporation in the aqueous phase that resulted in considerable, irreproducible inhibitory effects. This method showed that all *Lp* strains from clinical and environmental sources are sensitive to T-4-ol at 36 ± 1 °C. It is important to underline that the T-4-ol and TTO MBCs were equal, highlighting that the main component of the mixture responsible for the antimicrobial activity was T-4-ol, confirming previous research [39,42,43,47,48].

According to some authors [47,59] the increased antimicrobial activity of T-4-ol depends on the simultaneous presence of both hydrophilic and hydrophobic characteristics sufficient to allow diffusion through the water present around the bacterial cytoplasmic membrane and through the phospholipid layer of the cytoplasmic membrane [59]. While the T-4-ol present in the TTO mixture would have a reduced bactericidal activity level due to the presence of non-oxygenated components, such as gamma-Terpinene, responsible for reducing T-4-ol aqueous solubility and consequently the concentration of molecules of T-4-ol on the bacterial surface [60].

*Legionella* are remarkably resistance to high temperatures, having been isolated from hot water systems up to 66 °C, with a variability of growth depending on the temperature [61,62]. Studies have shown that *Lp* naturally multiplies in water at temperatures between 25 °C and 45 °C, with an optimal temperature range of 32–42 °C [63]. This bacterium can grow in both thermal waters [64] and in the hot water systems of hospitals, hotels, and private apartments [65,66]. The effects of temperature on T-4-ol and TTO activity against *Lp* of at 40 °C and 45 °C showed a decrease in both MIC and MBC values with increasing temperature for both substances. Specifically, the MIC and MBC values obtained by testing the T-4-ol activity were always lower than with the TTO, highlighting the stronger antimicrobial effect of T-4-ol. A temperature of 45 °C gave statistically significant MBC values when compared to those obtained at 36 °C and 40 ± 1 °C.

To ascertain the contribution that vapor makes on the inhibitory and killing activities of T-4-ol and TTO, the micro-atmosphere diffusion method at 36 °C and 45 ± 1 °C showed that the temperature does significantly influence the effectiveness of the vapor of both the TTO and T-4-ol. As expected, an increase in the evaporation occurred at higher temperatures for both the phytocomplex and the isolated component, although there was clear evidence that the anti-*Legionella* activity was seen withT-4-ol at 36 ± 1 °C. It could be hypothesized that T-4-ol is more effective than TTO at 36 °C and not at 45 ± 1 °C because of the greater volatilization of the various components of the TTO that act synergistically in the phytocomplex. The anti-*Lp* activity of T-4-ol, both in liquid and vapor phase, clearly showed that this main single component can exert antimicrobial activity greater than or similar to that of the TTO.

In vitro survival of *Lp* treated with 0.5% *v*/*v* T-4-ol (corresponding to the MBC value) at various contact times highlighted the *Lp* killing in over 50% of bacteria in 5 min, reaching 100% killing in 10 min. As mentioned above, previous research would therefore be confirmed indicating that T-4-ol is mainly responsible for the antimicrobial activity of TTO. Further studies are needed to clearly define the kinetics of T-4-ol in comparison with TTO and determine the optimal bactericidal concentration against *Lp*.

The antibacterial activity of EOs has not been related to a specific mechanism but there are descriptions of damage to the cytoplasmic membrane and the cell wall, with consequent leak of cellular content [59,67,68,69,70,71]. The effects of EOs on the *Lp* cell morphology have not been extensively investigated and to date, there is only one study on the action of the EO of *Thymus vulgaris*, chemotype carvacrol [36], using SEM and Transmission Electron Microscopy (TEM). This research showed morphological modifications of the *Legionella* surface when treated with this EO. *Lp* possesses a higher proportion of branched-chain fatty acids in the cell wall compared to other Gram-negative bacteria [72]. This unique cell wall composition could allow for the expression of changing cell morphologies when the organism is treated with different concentrations of EO and its components.

In this study, for the first time, the activity of TTO and T-4-ol on *Legionella* was analyzed by SEM. After *Lp* cell treatment with T-4-ol at concentrations from 0.25 to 1.0% *v*/*v*, at 30 °C for 1 h, SEM observations showed a detrimental effect on the cell surface morphology as a function of the concentration of T-4-ol, (micrographs not shown). At the lowest concentration (0.25%), a slight difference could be revealed between the T-4-ol-treated cells and the control. However, 1% *v*/*v* T-4-ol altered the cell surface, creating marked cell shrinkage and randomly distributed cytoplasmic expulsions.

We also evaluated the influence of different incubation temperatures, 40 °C and 45 °C, following TTO and T-4-ol treatment, inducing the appearance of wrinkled and ruffled structures on the bacterial surface. Furthermore, 0.42% of T-4-ol, the concentration present in TTO at 1% *v*/*v* and corresponding to the sub-MBC, at 40 °C and 45 °C, caused the formation of small bubbles, probably due to the selective folding of the external cell wall, while leaving the inner membrane unchanged (micrograph not shown).

In particular, the 1 h treatment with T-4-ol 1%, at 45 °C induced even more marked alterations, generating numerous flattened cells differing from those found on cell surfaces treated with TTO (Figure 5E). These effects could be caused by the lipophilic nature of T-4-ol [73]. It has been hypothesized that T-4-ol can spread and damage cell membrane structures, causing greater fluidity or altering membrane organization and inhibition of membrane-bound enzymes [74]. SEM observations showed that TTO or T-4-ol treatment combined with increasing temperature induced severe morphological and structural alterations in *Lp* cells, including collapse of the spheroplast-like structure and opening of the cell membrane envelope.

This data has confirmed the synergistic effect of increased temperature (45 °C) with T-4-ol or TTO on the reduction of *Lp* growth and has identified methods to improve the effectiveness of these natural products.

## 4. Materials and Methods

### 4.1. Terpinen-4-ol and Melaleuca alternifolia Cheel (Tea Tree) Essential Oil

T-4-ol (CAS number 562-74-3) was extracted by vapor distillation of Australian *Melaleuca alternifolia* (Maiden & Betch) Cheel essential oil [Tea tree oil (TTO)-Pharma Grade], both supplied by Variati (Milan, Italy).

TTO was analyzed, prior to use, for the determination of single constituents by gas chromatography (GC-FID) and gas-chromatography-mass spectrometry (GC-MS), while complying with the International Standard ISO 4730 [75,76] and the European Pharmacopeia [77] as previously reported [27,42]. T-4-ol and 1,8 cineole, at a concentration of 42.35% and 3.57%, respectively, were the typical constituents that characterize the standard TTO composition (T-4-ol type).

### 4.2. Micro-Organisms and Culture Media

Twenty-one *Lp* clinical and environmental isolates from our frozen stock collection were tested. They included: American Type Culture Collection (ATCC) strains-*Lp* sg1 (*Lp*1) ATCC 33152 and *Lp* sg6 (*Lp*6) ATCC 33215, five *Lp*1 clinical isolates, five *Lp*1 environmental isolates, five *Lp*6 clinical isolates, and four *Lp*6 environmental isolates. *Escherichia coli* ATCC 25922 was used as quality control. The storage and culture conditions of all bacterial strains used to perform the micro-dilution and micro-atmosphere diffusion methods with T-4-ol were already described for TTO experiments [27]. Six selected *Lp* strains sg1 (one ATCC 33152 reference strain, one clinical, and one environmental strains) and sg6 (one ATCC 33215 reference strain, one clinical, and one environmental strains) were tested by both the broth micro-dilution and micro-atmosphere diffusion methods at various temperatures.

### 4.3. Determination of the Minimum Inhibitory (MIC) and Bactericidal (MBC) Concentrations by Broth Micro-Dilution Method

Susceptibility testing of T-4-ol and TTO were performed according to the Clinical and Laboratory Standards Institute (CLSI) broth micro-dilution method, with some modifications as previously reported [27]. T-4-ol and TTO were diluted using Buffered Yeast Extract Broth (BYEB) (Oxoid™, Thermo Fisher Diagnostics Limited, Cheshire, UK) in the presence of Tween™ 80 at 0.001% *v*/*v*. Experiments were performed to evaluate the minimal Tween 80 concentration necessary to solubilize T-4-ol and TTO and able to maintain *Lp* viability. These were carried out using BYEB at 36 ± 1 °C for 72 h, as previously described [27]. Aliquots of 50 µL of two-fold dilutions of T-4-ol or TTO solutions were dispensed in 96-well microtiter plates. The final concentration of T-4-ol and TTO ranged from 0.0078% to 4% *v*/*v*. For the inoculum, 72 h agar cultures of *Lp* were suspended in sterile distilled water to optical density (OD = 0.6) measured at 600 nm (≅10^9^ CFU/mL), vortex-mixed and diluted to give ≅10^8^ CFU/mL. Fifty microliters of each diluted suspension (≅10^8^ CFU/mL) were added in duplicate to two-fold dilutions of TTO- or T-4-ol- solutions to give final concentration of ≅2.5 × 10^6^ cells/well. Microtiter plates were then incubated at 36 ± 1 °C with 2.5% CO_2_ for 72 h covered with a sterilized Transparent Microplate Sealer (TMS) (AMP Llseal, Greiner Bio-one, Frickenhausen, Germany). The control of bacterial growth was performed in T-4-ol and TTO free-medium added with 0.001% *v*/*v* Tween 80. Each tray included a sterility column (eight wells) without inoculum.

The MIC values were defined as the lowest T-4-ol or TTO concentrations, showing a growth inhibition of 99% compared with the oil-free *Lp* control growth determined either spectrophotometrically (450 nm) or visually. The MBC values were determined by plating 10 µL from each well with no apparent growth onto Buffered Charcoal Yeast Extract with 0.1% alpha-ketoglutarate (BCYE-α, Oxoid™, Thermo Fisher Diagnostics Limited, Cheshire, UK) and incubated at 36 ± 1 °C with 2.5% CO_2_. After 72 h, the viability was assessed. Each experiment was carried out in duplicate. MBC were defined as the lowest oil concentration resulting in the death of 99.9% of the initial inoculum.

To test the influence of temperature, on T-4-ol and TTO antimicrobial activities, six representative *Lp* strains sg1 and sg6 were tested at three different temperatures (36 °C, 40 °C, 45 ± 1 °C) (see Table 1).

### 4.4. Micro-Atmosphere Diffusion Method

The same six selected strains tested with the broth micro-dilution method at different temperatures were also tested with a slightly modified agar diffusion method to estimate the T-4-ol and TTO antimicrobial activities in vapor phase [78,79,80] as described in a previous report [27].

The BCYE-α agar medium was inoculated with ≅1 × 10^8^ CFU/mL *Lp* suspension and allowed to dry. A micro-coverglass (Prestige, Italy, 22 mm × 22 mm) was attached with a drop of commercial enamel on the upper lid of a Petri dish. Subsequently, a paper disc 6 mm in diameter (Oxoid, Unipath LTD, Basingstoke, England), moistened with 10 µL of pure T-4-ol or TTO, was put at the center of this slide. The surface of the disc was at a distance of about 4 mm from the growth surface of the test organism. The upper lid was then inverted, sealed with two sheets of parafilm (Pechiney Plastic Packaging, Chicago, Illinois, USA) to prevent vapor leakage. Plates were incubated at 36 °C and 45 ± 1 °C in 2.5% CO_2_ for 48 h and observed after 7 days to check the diameter of the resulting inhibition zone. BCYE-α agar with Tween 80 (0.01%) in absence of oil, was used as positive growth control. The space inside the sealed Petri dish was calculated to be ~60 cm^3^ of air. Each test was performed in duplicate and the mean values of the growth inhibition zone were determined after 7 days.

### 4.5. Time-Killing Assay

The in vitro survival of the reference strain *Lp* 1 ATCC 33152 as a function of T-4-ol concentration at 0.5% *v*/*v* and different contact times (3, 5, 10, 15, 30 min) was investigated at 30 °C, according to the EN 13623:2020 norm, slightly modified [81].

The strain stored at −80 °C in skimmed milk was thawed, cultured on BCYE-α agar, and incubated at 36 ± 1 °C in a humidified incubator with 2.5 % CO_2_ for 72 h. At the end of incubation, the strain was sub-cultured under the same conditions. The confluent grown subculture was the working culture. In order to prepare the bacterial suspension, the working culture was removed from the plate by suspending in Page’s saline to a concentration of ≅1 × 10^9^ CFU/ mL (OD_600_ = 0.6), ten-fold more concentrated as 1/10 dilution performed in the test.

Hard water (6 mL of solution A, consisting of 19.84 g of MgCl_2_ and 46.24 g of CaCl_2_ per liter and 8 mL of solution B, containing 35.02 g of NaHCO3 per liter, pH 7 ± 2) was used as the aqueous media in the test. Experiments were performed to a final inoculum concentration of ≅1 × 10^8^ CFU/mL at 30 °C (test suspension). The inoculum concentration at time zero was determined by CFU enumeration in BCYE-α agar medium. One mL of the bacterial test suspension was added to 8.9 mL of hard water plus 0.1 mL of yeast extract solution (as interfering substance) with the addition of Tween 80 (0.001% *v*/*v*) and T-4-ol at final concentration of 0.5% *v*/*v*. The concentration of 0.5% *v*/*v* T-4-ol was selected as it corresponds to the anti *Lp* MBC.

Briefly, the test solution (hard water plus the interfering substance) was added into a sterile screw cap glass tube and added to a shaker water bath at the 30 °C for 15 min. At the same time, the *Lp* test suspension was also maintained in a water bath at 30 °C. At the beginning of the contact time, the *Lp* test suspension was added to the test solution, mixed, and timing was immediately started for the chosen contact time. Shortly before the end of each contact time, the tube was mixed and 1 mL of the suspension taken and immediately diluted with 9 mL of hard water, then serially diluted ten-fold and 0.2 mL of each dilution, plated in duplicate on BCYE-α agar plates. Plates were incubated at 37 °C in 2.5% CO_2_ and colonies enumerated, after 7–10 days, to determine survival. Each test was carried out in duplicate.

### 4.6. Electron Microscope Analysis

#### 4.6.1. *Legionella pneumophila* Inoculum

*Lp* ATCC 33152 sg1 was selected as the test organism. Storage and culture of the strain for electron microscope analysis are previously described [27]. The seeded plates with *Lp* ATCC 33152 were incubated for 48–72 h at in 2.5% CO_2_. A small quantity of bacterial growth (about 10 µL) was harvested from plates with a platinum loop, added to 50 mL of BYEB and incubated overnight, with gentle shaking at 36 ± 1 °C. The culture was transferred into 500 mL of BYEB, incubated again overnight until reaching an OD of 2.8 at 600 nm. The cell suspension was centrifuged (20 min at 3500× *g*) to obtain the bacterial pellet. A bacterial suspension of ≅10^11^ CFU/mL was prepared in sterile distilled water and transferred into 9 mL tubes with test conditions (with defined % *v*/*v* TTO or T-4-ol, (see Section 4.6.2) to a final concentration of ≅10^10^ cells/mL. Each experiment was performed in duplicate.

#### 4.6.2. Scanning Electron Microscopy (SEM)

The cytological effects of T-4-ol and TTO on *Lp* ATCC 33152 sg1 (≅10^10^/mL) were observed by SEM analysis after one hour contact time at various temperatures.

The concentrations of T-4-ol (0.25%, 0.5%, 1% *v*/*v*) were tested at 30 °C, while only the concentrations of T-4-ol (0.42% *v*/*v* and 1% *v*/*v* = 2 MBC) and TTO (1% *v*/*v* = 2 MBC and 2.5% *v*/*v*) were tested at 40 °C and 45 °C. *Lp* suspension without TTO or T-4-ol at 0 min and 60 min at 30 °C, 40 °C, and 45 °C were used as controls to verify CFU/mL concentration by plating, after serial tenfold dilutions of the bacterial suspension onto BCYE and incubating at 36 ± 1 °C in 2.5% CO_2_ for 72 h (Appendix A).

Serial ten-fold dilutions of the initial (at time 0, control) and final (after 1 h) test conditions after T-4-ol or TTO addition, were performed to confirm the viability and number of CFU/mL by plating onto BCYE agar and incubating at 36 ± 1 °C in 2.5% CO_2_ for 72 h. At the end of the contact times, the mixtures with gentle shaking were transferred into 50 mL tubes, and centrifuged (20 min at 3500 g) to obtain the bacterial pellets discarding the supernatant. After two washes with sterile distilled water, the quantity of bacterial pellet required for microscopic examination was taken with a sterile loop and fixed for 30 min at room temperature in 2.5% (*v*/*v*) glutaraldehyde in 0.2 M cacodylate buffer (pH 7.4). After three washes in the same buffer, the bacterial cells were post-fixed with 1% (*w*/*v*) Osmium tetroxide (OsO_4_) for 1 h, dehydrated on an ethanol gradient, critical point dried in CO_2,_ and gold coated by sputtering. The samples were examined with a Field Emission Gun Scanning Electron Microscope (FEG-SEM) (FEI, Eindhoven, the Netherlands).

### 4.7. Statistical Analysis

Normal distribution data were analyzed using mean and standard deviation parameters. Two-way ANOVA test was used to analyze data obtained from the broth micro-dilution method. Fisher’s LSD test was used to analyze data obtained from the micro-atmosphere diffusion test. One-way t-test was used to assess the effectiveness of 0.5% *v*/*v* of T-4-ol on cell viability. GraphPad PRISM version 8 3.0 XLM software (Dotmatics, Boston, MA, USA) as used to develop the statistical tests.

## 5. Conclusions

We have shown that *Lp*, irrespective of serogroup and source of isolation, were sensitive to T-4-ol, the main compound of the TTO mixture, and the effect was bactericidal. We also showed that the T-4-ol vapor and temperature exerted critical activity on the *Lp* cells. Moreover, our data confirmed previous hypotheses that T-4-ol was the main component responsible for the microbicidal activity of TTO. As TTO contains a lot of components, each with individual properties, it would be more practical to make a formulation containing a single active ingredient as a potential alternative or supplemental agent for the control of *Legionella* in water systems. T-4-ol could rightly be included in the list of potential biocides active against *Legionella*, used in combination with other chemical compounds or physical treatments for the control of *Legionella* contamination, especially in small waterlines or in particularly respiratory medical devices. Another important application could be spa pools, which represent an important risk of legionellosis due to aerosol inhalation, as documented by the legionellosis outbreaks associated with spa pools [82]. Further studies are needed to calculate the effective anti-*Lp* dose of the TTO main component on the spot and its possible toxicity. This study suggests that T-4-ol, both in liquid and vapor phase, could be considered a promising and innovative method for the control of *Legionella* in water systems.

## 6. Patents

There is a patent of Francesca Mondello and Maria Luisa Ricci (International application No. PCT/IT2011/000267; publication number WO/2012/014244A1) resulting from the work reported in this manuscript. It refers to the use of T-4-ol as an antimicrobial agent against bacteria of the genus *Legionella*, for the disinfection of distribution water systems, cooling towers, spas, small waterlines such as a dental units, and medical devices such as respirator medical devices [83].

## Figures and Tables

**Figure 1 pathogens-11-00682-f001:**
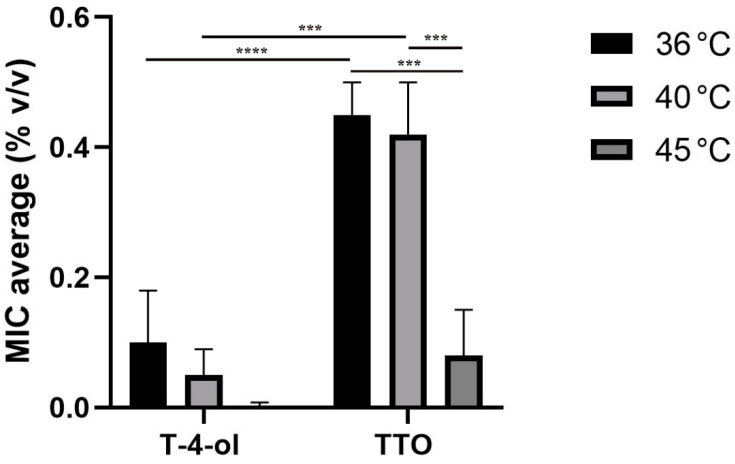
Comparison of inhibitory activities of terpinen-4-ol and tea tree oil against six *Legionella pneumophila* serotypes 1 and 6 with broth micro-dilution method and sealing of microtiter wells at various temperatures, (*** *p* < 0.0002, **** *p* < 0.0001).

**Figure 2 pathogens-11-00682-f002:**
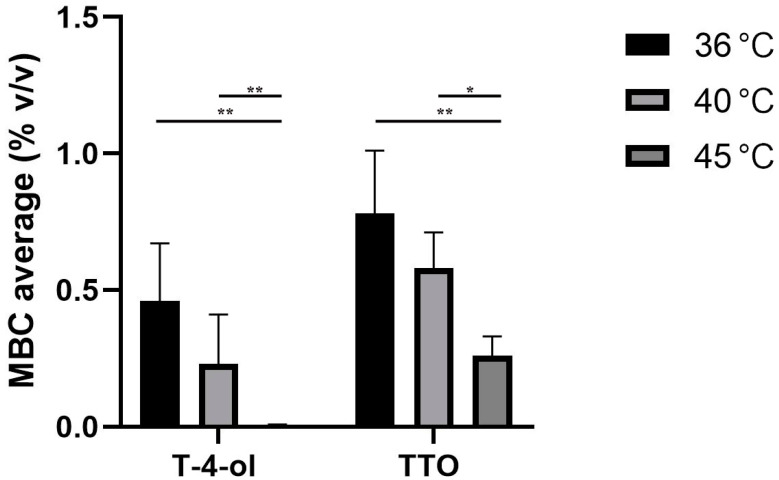
Comparison of cytocidal activities of terpinen-4-ol and tea tree oil against six *Legionella pneumophila* serogroups 1 and 6 with broth micro-dilution method and sealing of microtiter wells at various temperatures, (* *p* < 0.03, ** *p* < 0.002).

**Figure 3 pathogens-11-00682-f003:**
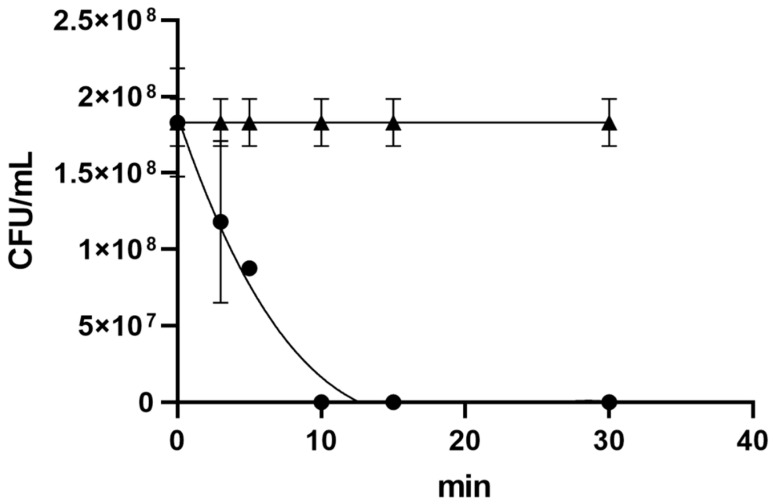
*Legionella pneumophila* CFU/mL as a function of the terpinen-4-ol concentration (0.5% *v*/*v*) and contact times (dots). *L. pneumophila* CFU/mL without terpinen-4-ol (triangles).

**Figure 4 pathogens-11-00682-f004:**
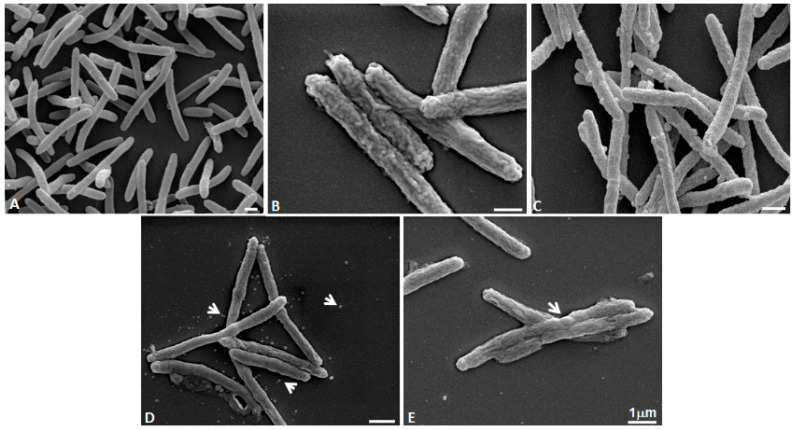
SEM observations of *Legionella pneumophila* exposed at 40 °C untreated (**A**) and treated with different concentrations of terpinen-4-ol (T-4-ol) and tea tree oil (TTO). (**B**) 1% *v*/*v* TTO; (**C**,**D**) 2.5% *v*/*v* TTO; (**E**) 1% *v*/*v* T-4-ol. Arrows indicate blebs.

**Figure 5 pathogens-11-00682-f005:**
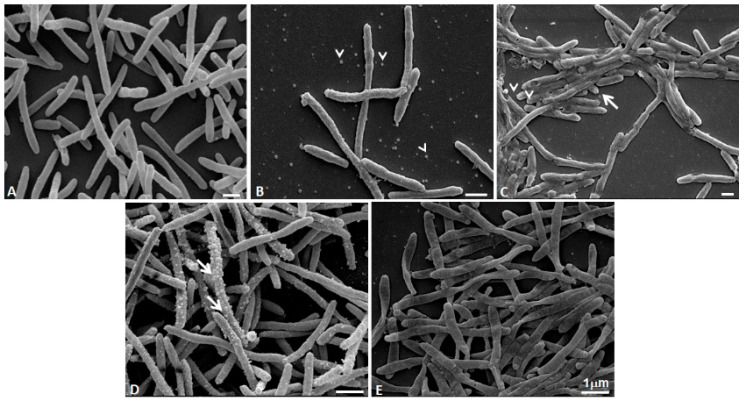
SEM observations of *Legionella pneumophila* exposed at 45°C untreated (**A**) and treated with different concentrations of terpinen-4-ol (T-4-ol) and tea tree oil (TTO). (**B**,**C**) 1% *v*/*v* TTO; (**D**) 2.5% *v*/*v* TTO; (**E**) 1% *v*/*v* T-4-ol. Arrows indicate blebs and damage on cell surface as well as collapsed bacteria.

**Table 1 pathogens-11-00682-t001:** Activity of terpinen-4-ol and tea tree oil against individual *Legionella pneumophila* (*Lp*) strains with micro-dilution method and sealing of microtiter wells at 36 ± 1 °C.

Tested Organisms	Terpinen-4-ol		Tea Tree Oil	
	MIC % (*v*/*v*)	MBC % (*v*/*v*)	MIC % (*v*/*v*)	MBC % (*v*/*v*)
*Lp*1 ATCC 33152 *	0.125	0.5	0.25	0.5
*Lp*1 1/3224 ^1,^*	0.125	0.5	0.25	1.0
*Lp*1 2510 ^1^	0.06	0.5	0.25	0.5
*Lp*1 41/2883 ^1^	0.125	0.5	0.25	0.5
*Lp*1 3260 ^1^	0.125	0.5	0.125	0.5
*Lp*1 4357 ^1^	0.125	0.5	0.25	0.5
*Lp*1 73 ^2^	0.125	0.5	0.25	0.5
*Lp*1 66 ^2,^*	0.125	0.5	0.25	0.5
*Lp*1 55/3646 ^2^	0.125	0.5	0.25	1.0
*Lp*1 2258 ^2^	0.125	0.5	0.25	0.5
*Lp*1 3261 ^2^	0.125	0.5	0.25	0.5
*Lp*6 ATCC 33215 *	0.125	0.5	0.25	0.5
*Lp*6 11/2378 ^1,^*	0.125	0.5	0.5	0.5
*Lp*6 51/3380 ^1^	0.125	0.5	0.25	0.5
*Lp*6 3265 ^1^	0.125	0.5	0.25	0.5
*Lp*6 3374 ^1^	0.125	0.5	0.25	0.5
*Lp*6 3303 ^1^	0.25	0.5	0.25	0.5
*Lp*6 44/2944 ^2^	0.125	0.5	0.25	0.5
*Lp*6 54/3645 ^2^	0.125	0.5	0.5	0.5
*Lp*6 2868 ^2^	0.125	0.5	0.25	0.5
*Lp*6 4405 ^2,^*	0.125	0.25	0.25	0.25

^1^ Clinical isolate; ^2^ Environmental isolate; ATCC = Reference strain from American Type Culture Collection (ATCC); * Strains used for broth micro-dilution and micro-atmosphere diffusion methods at various temperatures.

**Table 2 pathogens-11-00682-t002:** Comparison of the activities of terpinen-4-ol and tea tree oil against *Legionella pneumophila* (*Lp*) isolates with micro-dilution method and sealing of microtiter wells.

Tested Strains (*n*)	Test Agent	MIC % *v*/*v*		MBC % *v*/*v*	
		Range	MIC_90_	Range	MBC_90_
*Lp*1 ^§^ (11)	T-4-ol	0.06–0.125	0.125	0.5	0.5
	TTO	0.125–0.25	0.25	0.5–1.0	0.5
*Lp*6 ^§§^ (10)	T-4-ol	0.125–0.25	0.125	0.25–0.5	0.5
	TTO	0.25–0.5	0.25	0.5–1.0	0.5

^§^ (1 ATCC 33152 *Lp* 1 reference strain + 5 clinical and 5 environmental strains); ^§§^ (1 ATCC 33215 *Lp* 6 reference strain + 5 clinical and 4 environmental strains); *n*, number; MIC_90_ = MIC which inhibits 90% of isolates.

**Table 3 pathogens-11-00682-t003:** Micro-atmosphere diffusion method: terpinen-4-ol and tea tree oil vapors effects on *Legionella pneumophila* (*Lp*) after 7 days contact at different temperatures.

Strains	36 ± 1 °C			45 ± 1 °C		
	T-4-ol ^1^	TTO ^1^	T-4-ol/TTO	T-4-ol ^1^	TTO ^1^	T-4-ol/TTO
*Lp* *1 ATCC 33152*	90	20	-	NG	NG	-
*Lp* *1 66* ^2^	48	17	2.82	77	40	1.93
*Lp* *1 3224* ^3^	30	15	2.00	90	90	1.00
*Lp* *6 ATCC 33215*	20	15	1.33	90	90	1.00
*Lp* *6 4405* ^2^	48	20	2.40	90	38	2.37
*Lp**6 11*/*2378*^3^	90	25	3.60	90	90	1.00
*AV ± SD*	47.20 ± 24.0	18.40 ± 3.8	2.43 ± 0.8	87.4 ± 5.2	69.60 ± 25	1.46 ± 0.6

^1^ Inhibition area in mm at different temperatures; ^2^ Environmental isolate; ^3^ Clinical isolate. NG= no growth of this *Legionella* strain at 45 °C. Data are the average of three experiments. AV ± SD = Average + Standard Deviation.

## Data Availability

The data presented in this study are available on reasonable request addressed to the corresponding author.

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
