# Peer review of "Terpinen-4-ol, the Main Bioactive Component of Tea Tree Oil, as an Innovative Antimicrobial Agent against Legionella pneumophila"

_pathogens, 2022, doi:10.3390/pathogens11060682_

Round 1

Reviewer 1 Report

Manuscript "Terpinen-4-ol, the main bioactive component of tea tree oil, as an innovative antimicrobial agent against Legionella pneumophila" presents interesting results of research on the antimicrobial properties of natural substances. Due to the increasing phenomenon of drug resistance, the search for new substances with antimicrobial properties is very beneficial.

Detailed comments:

Too many keywords, please choose the most important ones.

Table 3 - how many repetitions were made? There is no statistical analysis of the results.

Figure 3 - it would be better to give the results on a semi-log scale. The determined curve must not show results below zero.

line 499 - names must be in italics.

Author Response

REVIEWER 1

Thank your precious comments/remarks to our Manuscript 1744154:  "Terpinen-4-ol, the main bioactive component of tea tree oil, as an innovative antimicrobial agent against Legionella pneumophila".

 Please find below the responses (R):

Comments and Suggestions for Authors

Manuscript "Terpinen-4-ol, the main bioactive component of tea tree oil, as an innovative antimicrobial agent against Legionella pneumophila" presents interesting results of research on the antimicrobial properties of natural substances. Due to the increasing phenomenon of drug resistance, the search for new substances with antimicrobial properties is very beneficial.

Detailed comments:

1.Too many keywords, please choose the most important ones.

  1. The most important keywords have been chosen as suggested by the journal’s instructions for Authors: “Keywords:Three to ten pertinent keywords need to be added after the abstract. We recommend that the keywords are specific to the article, yet reasonably common within the subject discipline”.
  2. Table 3 - how many repetitions were made? There is no statistical analysis of the results. R.As already specified in legend, data are the average of three experiments and each test was done at least in duplicate (see Materials and Methods, line 46).Furthermore, Lines 178-183 show the statistical data calculated according to Fisher test.
  1. Figure 3 - it would be better to give the results on a semi-log scale. The determined curve must not show results below zero. R. We prefer not to transform the Y axis to semi-log scale. The graph has been changed in the text, now no data is less than zero.
  2. line 499 - names must be in italics. R.Names have been changed in italics as suggested.

Reviewer 2 Report

In the manuscript entitled “Terpinen-4-ol, the main bioactive component of tea tree oil, as an innovative antimicrobial agent against Legionella pneumophila” the authors have investigated and compared the in vitro antimicrobial activity of Terpinen-4-ol and tea tree essential oil against different strains of L. pneumophila.

The lack of studies in this regard and considering the relevant Legionella issue, I suggest the editor to accept the publication of the article with minor revisions. The article is well written and the English only requires some minor revision.

I have the impression that some commas are missing throughout the text. Please, check.

ABSTRACT

Slightly exceeds the maximum number of words allowed (200). Improve

INTRODUCTION

Overall, the introduction is well written and only minor revisions are needed. The background of the study is quite clear as well as the aims.

Line 61: Add reference/s.

Line 74: In this sentence, the verb seems to be missing. Please, check.

Line 81: This is the first time in the main text that the abbreviation TTO is used. Please write the full name.

Line 83: This is the first time in the main text that the abbreviation T-4-ol is used. Please write the full name.

Line 90: Check the word “effectiveness”.

RESULTS

In general, the results are sufficiently presented even if some revisions are necessary, especially in the tables and figures which I consider approximate and sometimes not entirely clear. In fact, some notes lack essential information or are partially presented.

Lines 105-107: Arrange the notes of the tables with respect to the text are joined together.

Line 109: This is the first time in the main text that Table 2 appeared. Although it is clear from the text what Table 2 refers to, it is not correct that the reader should deduce it. Then arrange the Table 2 so that it is clear what results it refers to.

Lines 127-132: The formatting of Table 2, 3 is approximative. Please, fix it.

Line 112-114: I don't understand what the authors mean. What does it mean they were considered a whole? The reader has no idea what is written next so it would be good to improve this part.

Line 134: Only here, I understand that the analyzes described above were carried out at different temperatures. Therefore, I suggest the authors to specify in the text at what temperatures the results of the MIC and MBC presented above (Lines 100-102) were obtained.

Line 174: “…in accordance with literature data…”, please, add reference.

Lines 180-183: I suggest moving this sentence to the beginning of the paragraph since it is the most important result this paragraph is based on.

Line 194: “We highlighted the strong bactericidal action of T-4-ol against Lp.” Too generic. strong might mean nothing. gets better. Improve

Line 195: “…in the presence of a concentration of…” I suggest simplifying with “at”.

In the notes of the tables and figures, specify how many repetitions were performed for each analysis.

Paragraph 2.4. I do not like that there is information that should be specified in the notes of the figures (arrows). Improve

Figure 3: fix the legend of the graph (they are different).

Figure 4: improve notes. it is difficult to understand what the letters refer to. Furthermore, specify what the arrows refer to.

DISCUSSIONS

Overall, the discussions are well presented and in line with the results. In my opinion, no particular reviews are needed.

Lines 244-246: This sentence should be written better. Looks like verb is missing.

Line 247: write the genus (M. alterniflora), it is written here for the first time.

Line 272: “Van del Vel…” the reference is missing.

MATERIALS AND METHODS

Line 430: Please, specify in detail the growth conditions of Legionella (media, times and temperatures).

Line 533: Please, specify the type of test used to evaluate the normal distribution of the data.

Author Response

REVIEWER 2

Thank your precious comments/remarks to our Manuscript 1744154:  "Terpinen-4-ol, the main bioactive component of tea tree oil, as an innovative antimicrobial agent against Legionella pneumophila".

 Please find below the responses (R):

Comments and Suggestions for Authors

In the manuscript entitled “Terpinen-4-ol, the main bioactive component of tea tree oil, as an innovative antimicrobial agent against Legionella pneumophila” the authors have investigated and compared the in vitro antimicrobial activity of Terpinen-4-ol and tea tree essential oil against different strains of L. pneumophila.

The lack of studies in this regard and considering the relevant Legionella issue, I suggest the editor to accept the publication of the article with minor revisions. The article is well written and the English only requires some minor revision.

I have the impression that some commas are missing throughout the text. Please, check.

R.The text has been checked and some commas have been added.

ABSTRACT

Slightly exceeds the maximum number of words allowed (200). Improve

R. The abstract contains less than 200 words (i.e. 195), according to the instruction for Authors.

INTRODUCTION

Overall, the introduction is well written and only minor revisions are needed. The background of the study is quite clear as well as the aims.

Line 61: Add reference/s.

R. The reference has been added as suggested

Line 74: In this sentence, the verb seems to be missing. Please, check.

R. The sentence has been changed as follows:

The biological activities of EOs are considered to have preventive and therapeutic benefits [19-21].

Line 81: This is the first time in the main text that the abbreviation TTO is used. Please write the full name.

R. The full name has been added in the text as suggested.

Line 83: This is the first time in the main text that the abbreviation T-4-ol is used. Please write the full name.

R. The full name has been added in the text as suggested.

Line 90: Check the word “effectiveness”.

R. The sentence has been changed as follows:

This is the first report that highlights the in vitro susceptibility of Lp to T-4-ol in comparison to the anti-Lp activity of TTO with the aim to develop new disinfectants for Legionella contaminated water distribution systems.

RESULTS

In general, the results are sufficiently presented even if some revisions are necessary, especially in the tables and figures which I consider approximate and sometimes not entirely clear. In fact, some notes lack essential information or are partially presented.

Lines 105-107: Arrange the notes of the tables with respect to the text are joined together.    

R. Amended.

Line 109: This is the first time in the main text that Table 2 appeared. Although it is clear from the text what Table 2 refers to, it is not correct that the reader should deduce it. Then arrange the Table 2 so that it is clear what results it refers to.

R. The sentence has been modified as follows:

The MIC and MBC average values of T-4-ol and TTO against the two Lp serogroups were compared (Table2). The uniformity of MIC values at 0.25% v/v for TTO and at 0.125% v/v for T-4-ol are noticeable, with rare exceptions, while the MBC values at 0.5% v/v was the same for both TTO and for the main component T-4-ol in both tables (Tables 1 and Table 2).

Lines 127-132: The formatting of Table 2, 3 is approximative. Please, fix it.

R. Amended

Line 112-114: I don't understand what the authors mean. What does it mean they were considered a whole? The reader has no idea what is written next so it would be good to improve this part.

R. The sentence was modified as follows:

No difference either between the two serogroups or among the reference strains environmental and clinical isolates was seen, and for this reason the strains were grouped according to their serogroup in Table 2.

Line 134: Only here, I understand that the analyzes described above were carried out at different temperatures. Therefore, I suggest the authors to specify in the text at what temperatures the results of the MIC and MBC presented above (Lines 100-102) were obtained.

R. Line 97: In the text has been specified at which temperatures the MIC and MBC results were obtained in table 1 and table 2 as recommended.

Line 174: “…in accordance with literature data…”, please, add reference.

R. The reference 29 was already present at the end of the sentence. It has been changed as follows:

The Lp sg1 ATCC 33152 strain did not grow at 45 °C therefore it was not considered for experiments at that temperature [29].

Lines 180-183: I suggest moving this sentence to the beginning of the paragraph since it is the most important result this paragraph is based on.

R. As suggested, the sentence was moved at the beginning of the paragraph.

Line 194:We highlighted the strong bactericidal action of T-4-ol against Lp.” Too generic. strong might mean nothing. gets better. Improve

R. The text was modified as suggested: We highlighted the exponential trend of T-4-ol bactericidal activity against Lp.

Line 195: “…in the presence of a concentration of…” I suggest simplifying with “at”.

R. The sentence was modified as suggested.

In the notes of the tables and figures, specify how many repetitions were performed for each analysis.

R. We disagree. In our opinion it is enough to specify in the Materials and Methods paragraph.

Paragraph 2.4. I do not like that there is information that should be specified in the notes of the figures (arrows). Improve

R. The suggestion has been considered. 

Figure 3: fix the legend of the graph (they are different).

R. Amended

Figure 4: improve notes. it is difficult to understand what the letters refer to. Furthermore, specify what the arrows refer to.

R. The text and the notes were modified as follow:

Lines 217-218

Moreover, numerous blebs were present on the coverslips (Figure 4D). Figure 4E shows that treatment with 1% v/v T-4-ol induced the swelling and wrinkling of the cell surface.

Lines 222-224

 Figure 4.  SEM observations of Legionella pneumophila exposed at 40°C untreated (A) and treated with different concentrations of terpinen-4-ol (T-4-ol) and tea tree oil (TTO). B: 1% v/v TTO; C - D: 2.5% v/v TTO; E: 1% v/v T-4-ol. Arrows indicate blebs.

Lines 241-243

Figure 5. SEM observations of Legionella pneumophila exposed at 45°C untreated (A) and treated with different concentrations of terpinen-4-ol (T-4-ol) and tea tree oil (TTO). B-C: 1% v/v TTO; D: 2.5% v/v TTO; E: 1% v/v T-4-ol. Arrows indicate blebs and damage on cell surface as well as collapsed bacteria.

DISCUSSION

Overall, the discussions are well presented and in line with the results. In my opinion, no particular reviews are needed.

Lines 244-246: This sentence should be written better. Looks like verb is missing.

R. The sentence was modified as suggested

Line 247: write the genus (M. alterniflora), it is written here for the first time.

R.The genus was added as suggested

Line 272: “Van del Vel…” the reference is missing.

R.The reference was added as suggested

MATERIALS AND METHODS

Line 427: Please, specify in detail the growth conditions of Legionella (media, times and temperatures).

R.Considering that the reference relating to the method has been inserted, the text has been slightly modified as follows: Experiments performed to evaluate the minimal Tween 80 concentration necessary to solubilise T-4-ol and TTO able to maintain Lp viability were carried out using BYEB at 36 ± 1 °C for 72h as previously described [27].

Line 533: Please, specify the type of test used to evaluate the normal distribution of the data.

R. Line 538. The text was integrated as follows: “Normal distribution data were analyzed using mean and standard deviation parameters.”